[Supplementary Material]

# A  Theoretical Results

Consider a rewardless MDP $\langle \mathcal{S}, \mathcal{A}, T, \gamma \rangle$. Reward functions $R \in \mathbb{R}^{\mathcal{S}}$ have corresponding optimal value functions $V_R^*(s)$.

**Proposition 1** (Communicability bounds maximum change in optimal value)**.** *If $s$ can reach $s'$ with probability 1 in $k_1$ steps and $s'$ can reach $s$ with probability 1 in $k_2$ steps, then* $\sup_{R \in [0,1]^{\mathcal{S}}} \left| V_R^*(s) - V_R^*(s') \right| \leq \frac{1 - \gamma^{\max(k_1, k_2)}}{1 - \gamma} < \frac{1}{1 - \gamma}.$

*Proof.* We first bound the maximum increase.

$$\sup_{R \in [0,1]^{\mathcal{S}}} V_R^*(s') - V_R^*(s) \leq \sup_{R \in [0,1]^{\mathcal{S}}} V_R^*(s') - \left( 0 \cdot \frac{1 - \gamma^{k_1}}{1 - \gamma} + \gamma^{k_1} V_R^*(s') \right) \tag{2}$$

$$\leq \frac{1}{1 - \gamma} - \left( 0 \cdot \frac{1 - \gamma^{k_1}}{1 - \gamma} + \gamma^{k_1} \frac{1}{1 - \gamma} \right) \tag{3}$$

$$= \frac{1 - \gamma^{k_1}}{1 - \gamma}. \tag{4}$$

Equation (2) holds because even if we make $R$ equal 0 for as many states as possible, $s'$ is still reachable from $s$. The case for maximum decrease is similar. $\square$

# B  Training Details

In section 5.1, we aggregated performance from 3 curricula with 5 seeds each, and 1 curriculum with 3 seeds.

We detail how we trained the AUP and AUP$_{\text{proj}}$ conditions. An algorithm describing the training process can be seen in algorithm 1.

## B.1  Auxiliary reward training

For the first phase of training, our goal is to learn $Q_{\text{aux}}$, allowing us to compute the AUP penalty in the second phase of training. Due to the size of the full SafeLife state ($350 \times 350 \times 3$), both conditions downsample the observations with average pooling and convert to intensity values.

Previously, Turner et al. [22] learned $Q_{\text{aux}}$ with tabular Q-learning. They used environments small enough such that reward could be assigned to each state. Because SafeLife environments are too large for tabular Q-learning, we demonstrated two methods for randomly generating an auxiliary reward function.

AUP We acquire a low-dimensional state representation by training a continuous Bernoulli variational autoencoder [12]. To train the CB-VAE, we collect a buffer of observations by acting randomly for $\frac{100,000}{N_{\text{env}}}$ steps in each of the $N_{\text{env}}$ environments. This gives us 100K total observations with an $N_{\text{env}}$-environment curriculum. We train the CB-VAE for 100 epochs, preserving the encoder $E$ for downstream auxiliary reward training.

For each auxiliary reward function, we draw a linear functional uniformly from $(0, 1)^Z$ to serve as our auxiliary reward function, where $Z$ is the dimension of the CB-VAE's latent space. The auxiliary reward for an observation is the composition of the linear functional with an observation's latent representation.

AUP$_{\text{proj}}$ Instead of using a CB-VAE, AUP$_{\text{proj}}$ simply downsamples the input observation. At the beginning of training, we generate a linear functional over the unit hypercube (with respect to the downsampled observation space). The auxiliary reward for an observation is the composition of the linear functional with the downsampled observation.

The auxiliary reward function is learned after it is generated. To learn $Q_{\text{aux}}$, we modify the value function in PPO to a Q-function. Our training algorithm for phase 1 only differs from PPO in how we calculate reward. We train each auxiliary reward function for 1M steps.

## B.2  AUP reward training

In phase 2, we train a new PPO agent on $R_{\text{AUP}}$ (eq. (1)) for the corresponding SafeLife task. Each step, the agent selects an action $a$ in state $s$ according to its policy $\pi_{\text{AUP}}$, and receives reward $R_{\text{AUP}}(s, a)$ from the environment. We compute $R_{\text{AUP}}(s, a)$ with respect to the learned Q-values $Q_{\text{aux}}(s, \varnothing)$ and $Q_{\text{aux}}(s, a)$.

The penalty term is modulated by the hyperparameter $\lambda$, which is linearly scaled from $10^{-3}$ to some final value $\lambda^*$ (default $10^{-1}$). Because $\lambda$ controls the relative influence of the penalty, linearly increasing $\lambda$ over time will prioritize primary task learning in early training and slowly encourage the agent to obtain the same reward while avoiding side effects. If $\lambda$ is too large – if side effects are too costly – the agent won't have time to adapt its current policy and will choose inaction ($\varnothing$) to escape the penalty. A careful $\lambda$ schedule helps induce a successful policy that avoids side effects.

---

**Algorithm 1:** AUP Training Algorithm

---

**Initialize** Exploration buffer $S$
**Initialize** CB-VAE $\mathcal{F}$ with encoder $E$, decoder $D$
**Initialize** Exploration buffer $S$
**Initialize** Auxiliary reward functions $\phi$
**Initialize** Auxiliary policy $\psi_{\text{aux}}$, AUP policy $\pi_{\text{AUP}}$
**Require** CB-VAE training epochs $T$
**Require** AUP penalty $\lambda$
**Require** Exploration buffer size $k$
**Require** Auxiliary model training steps $L$
**Require** AUP model training steps $N$
**Require** PPO update function PPO-Update
**Require** CB-VAE update function VAE-Update
**for** *Step $k = 1, \ldots K$* **do**
    Sample random action $a$
    $s \leftarrow \text{Act}(a)$
    $S = s \cup S$
**end**
**for** *Epoch $t = 1, \ldots T$* **do**
    Update-VAE $(\mathcal{F}, S)$
**end**
**for** *Step $i = 1, \ldots L + N$* **do**
    $s \leftarrow$ Starting state
    **for** *Step $l = 1, \ldots L$* **do**
        $a = \psi_{\text{aux}}(s)$
        $s' = \text{Act}(a)$
        $r = \phi \cdot E(s)$
        PPO-Update $(\psi_{\text{aux}}, s, a, r, s')$
        $s = s'$
    **end**
    $s \leftarrow$ Starting state
    **for** *Step $n = 1, \ldots N$* **do**
        $a = \pi_{\text{AUP}}(s)$
        $s', r = \text{Act}(a)$
        $r = r + R_{\text{AUP}}(\psi_{\text{aux}}, \pi_{\text{AUP}}, s, a, \lambda)$   (Equation (1))
        PPO-Update $(\pi_{\text{AUP}}, s, a, r, s')$
        $s = s'$
    **end**
**end**

---

# C   Hyperparameter Selection

Table 2 lists the hyperparameters used for all conditions, which generally match the default SafeLife settings. *Common* refers to those hyperparameters that are the same for each evaluated condition. *AUX* refers to hyperparameters that are used only when training on $R_{\text{AUX}}$, thus, it only pertains to AUP and AUP$_{\text{proj}}$. The conditions PPO and Naive use the *PPO* hyperparameters for the duration of

their training, while `AUP`, `AUP`$_{\text{proj}}$ use them when training with respect to $R_{\text{AUP}}$. *DQN* refers to the hyperparameters used to train the model for `DQN`.

| Hyperparameter | Value |
|---:|:---|
| *Common* | |
| Learning Rate | $3 \cdot 10^{-4}$ |
| Optimizer | Adam |
| Gamma ($\gamma$) | 0.97 |
| Lambda (`PPO`) | 0.95 |
| Lambda (`AUP`) | $10^{-3} \rightarrow 10^{-1}$ |
| Entropy Clip | 1.0 |
| Value Coefficient | 0.5 |
| Gradient Norm Clip | 5.0 |
| Clip Epsilon | 0.2 |
| *AUX* | |
| Entropy Coefficient | 0.01 |
| Training Steps | $1 \cdot 10^{6}$ |
| *AUP$_{proj}$* | |
| Lambda (`AUP`) | $10^{-3}$ |
| *PPO* | |
| Entropy Coefficient | 0.1 |
| *DQN* | |
| Minibatch Size | 64 |
| SGD Update Frequency | 16 |
| Target Network Update Frequency | $1 \cdot 10^{3}$ |
| Replay Buffer Capacity | $1 \cdot 10^{4}$ |
| Exploration Steps | $4 \cdot 10^{3}$ |
| *Policy* | |
| Number of Hidden Layers | 3 |
| Output Channels in Hidden Layers | $(32, 64, 64)$ |
| Nonlinearity | ReLU |
| *cb-vae* | |
| Learning Rate | $10^{-4}$ |
| Optimizer | Adam |
| Latent Space Dimension ($Z$) | 1 |
| Batch Size | 64 |
| Training Epochs | 50 |
| Epsilon | $10^{-5}$ |
| Number of Hidden Layers (encoder) | 6 |
| Number of Hidden Layers (decoder) | 5 |
| Hidden Layer Width (encoder) | $(512, 512, 256, 128, 128, 128)$ |
| Hidden Layer Width (decoder) | $(128, 256, 512, 512, \text{output})$ |
| Nonlinearity | ELU |

Table 2: Chosen hyperparameters.

# D   Compute Environment

For data collection, we only ran the experiments once. All experiments were performed on a combination of NVIDIA GTX 2080TI GPUs, as well as NVIDIA V100 GPUs. No individual experiment required more than 3GB of GPU memory. We did not run a 3-seed `DQN` curriculum for the experiments in section 5.1.

| Condition | GPU-hours per trial |
|:---:|:---:|
| PPO | 6 |
| DQN | 16 |
| AUP | 8 |
| AUP$_{\text{proj}}$ | 7.5 |
| Naive | 6 |

Table 3: Compute time for each condition.

The auxiliary reward functions were trained on down-sampled rendered game screens, while all other learning used the internal SafeLife state representation. Incidentally, table 3 shows that AUP's preprocessing turned out to be computationally expensive (compared to PPO's).

# E   Additional Data

Figure 6 plots episode length and fig. 7 plots auxiliary reward learning. Figure 8 and fig. 9 respectively plot reward/side effects and episode lengths for each AUP seed. Figure 10 and fig. 11 plot the same, averaged over each curriculum; these data suggest that AUP's performance is sensitive to the randomly generated curriculum of environments.

Figure 6: Smoothed episode length curves with shaded regions representing $\pm 1$ standard deviation. AUP and AUP$_{\text{proj}}$ begin training on the $R_{\text{AUP}}$ reward signal at steps 1.1M and 1M, respectively.

Figure 7: Reward curves for auxiliary reward functions with a $Z$-dimensional latent space. Shaded regions represent $\pm 1$ standard deviation. Auxiliary reward is not comparable across trials, so learning is expressed by the slope of the curves.

Figure 8: Smoothed learning curves for individual AUP seeds. AUP begins training on the $R_{\text{AUP}}$ reward signal at step 1.1M, marked by a dotted vertical line.

Figure 9: Smoothed episode length curves for individual AUP seeds.

Figure 10: Smoothed learning curves for AUP on its four curricula. AUP begins training on the $R_{\mathrm{AUP}}$ reward signal at step 1.1M, marked by a dotted vertical line.

Figure 11: Smoothed episode length curves for AUP on each of the four curricula.