[Reviews · NeurIPS 2020]

Review 1

Summary and Contributions: UPDATE AFTER DISCUSSION: score 7 --> 8 The rebuttal was strong, and I believe the revision will effectively address the issues raised by reviewers (except the novelty, which was not a concern of mine). This paper demonstrates the potential of Attainable Utility Preservation (AUP, [25]) to prevent side-effects in complex environments from the SafeLife [27] test suite. The authors compare PPO with and without different side-effects penalties, and find that preserving attainable utility for a single random reward function works well across 4 different SafeLife tasks.  Specifically, it matches or exceeds PPO's performance in terms of reward, while significantly limiting side-effects (which are measured according to an unobserved true reward function). The paper's contributions include: 1) The first demonstration of an effective side-effects penalty (that does not also decrease reward) in SafeLife or a comparably complicated setting. 2) The experimental finding that AUP using a single reward function can be effective, which is non-obvious, and makes AUP appear significantly more promising as a practical method.

Strengths: The topic and the paper's contributions are highly significant and novel.  Very little work has been done on side-effects and this paper can be considered ground-breaking, as it is the first one to demonstrate strong results on a non-toy task using a method that has no clear limitations. The only obvious limitation would be the need to choose the auxiliary reward functions carefully, which is why the effectiveness of using a single reward function trained via unsupervised learning is very important. The work is well-motivated, very clearly presented, and the experimental results are strong and should spur more research in this direction.

Weaknesses: In order from most-to-least important, the paper has the following four notable weaknesses. I only view 2 of these as crucial to address, but I strongly encourage the authors to address 1-3 in their revision regardless.  1) The description of experiments lacks some important details: - CRUCIAL: The evaluation is unclear.  Were agents evaluated on held-out environments from the same task?  Or on the N_env training environments? Either way seems fine, but it should be specified! - The environments are not described well enough.  What is the state/observation space?  What are the rewards?  It's possible to look this up in [27], but this paper should include that information as well. - There is no discussion of model selection / early stopping.   The motivation for the method is that the true side-effect score cannot be calculated (else it could be incorporated directly into the reward function).  So would the auxiliary penalty term of AUP serve as a good measurement of how well the learner is avoiding side-effects?   - Related to the above: what happens if training continues after 5M frames?  Is it possible for the agent to "overfit" the single auxiliary reward function and end up causing more side-effects after longer training? - How were the hyperparameters chosen?  Since section 5.2 indicates some sensitivity to hyperparameters, it could help contextualize and potentially strengthen results in the rest of the paper if it could be stated that the default hyperparameters were chosen before the search was performed, or only based on the search in 1 of the 4 tasks (and then applied successfully to the other 3 tasks without any further tuning). - CRUCIAL: 160-161: This is unclear, and from the current description, it sounds like AUP might have had an unfair advantage in terms of how much data from each environment is got to see.  Please clarify. - 131: "randomly explores" means actions are chosen at uniform random? - CRUCIAL: for AUP_proj, does training of the policy only commence after 1.1M frames (as for AUP)? It doesn't look like it from the plots. 2) The paper could do a better job of motivating a side-effects penalty that does not help avoid mistakes during learning.  While the authors are clear that this is not the goal of AUP, they do not discuss to what extent this is a limitation. 3) The related work section provides good summaries, but often fails to make the connection to the present work clear. 4) The experiments are a bit thin. - They could use more comparisons (e.g. to reachability).  - They could use another RL algorithm besides PPO. - Does the primary reward function work as the auxiliary reward function?  This would be a very interesting experiment to run I think. - What happens if you use multiple auxiliary reward functions?

Correctness: I believe the paper is correct.  I think it should be considered only as a proof of concept that AUP can scale to complicated environments with limited overhead, not as a thorough evaluation of AUP and competitors.  But this is already a very significant contribution, and the work as it stands does a good job of presenting that conclusion and its justification, which I believe is sound.

Clarity: Besides the lack of details in the experiments, the paper has a few minor issues of clarity. Overall it is remarkably clear and well written. A few notable exceptions: - I didn't understand why Theorem 3 was described as: "Average optimal value penalties are large", and didn't understand it's purpose in this location. - I also found the explanation on lines 206-210 unclear and unconvincing.  In particular, the sentence on 208-209: why do you believe that?  And there are two sources of variability that are being conflated in this sentence: which auxiliary reward function, and which specific action (that decreases power). - 131-139: I suggest starting line 134 and 138 with "for the next 1M frames", and "for the final 3.9M frames".  It wasn't immediately obvious that these were 3 steps that occur in sequence while the agent interacts with the environment. - 114-120: I gather that this is how [27] did it, and that you are doing the same.  It we be clearer if that was more explicit.

Relation to Prior Work: The related work section provides good summaries, but often fails to make the connection to the present work clear.   The authors should discuss reachability a bit more.  Reachability has the advantage of not depending on the choice of auxiliary set (\R).  This work both shows that the choice of auxiliary set seems to matter (compare AUP / AUP_proj), and also that a simple approach to choosing auxiliary rewards can work in practice.

Reproducibility: Yes

Additional Feedback: Questions: - Would a similar approximation of reachability be feasible in these tasks? - 100: is this assuming rewards are in [0,1]?  This should be clarified if so. - 104-105: does AUP penalize increases in power as well?  This should be emphasized if so.  An example where that occurs would be great! Suggestions: - include a smoothed version of the learning curves for easier comparison. - put the learning curves for the same task in the same row instead of vertically stacked. - 240-241 "specify the right reward function" <-- I think this cuts against your pitch for this work.  In my mind, part of the promise of a side-effects penalty is that the specification can be less perfect. - Mention empowerment (e.g. Mohamed and Rezende, 2015) in related work and/or at lines 101-103. - 42 capitalize "proposition" - 138: "the agent learns to optimize R_AUP [...]" - 155: "we conduct 3 trials for each (condition,task) pair" - 111: "game" --> "games" Comments: - abstract line 8 has a category error; I would say "AUP [...] leads an agent to complete the task and avoid side effects" - 149: this is the first time you mention a "curriculum".  It is unclear what it refers to.


Review 2

Summary and Contributions: The work concerns avoiding undesired side effects in RL, which can occur when reward functions are not fully specified. It provides some theoretical results about the previously introduced AUP penalty. It also suggests SafeLife as a valuable testbed for this area and shows experimental results for a handful of cases in SafeLife. The experimental results show one surprising result that a single randomly generated reward function works well.

Strengths: Side effects in RL is a topic relevant to the NeurIPS community. I don’t find the results especially novel. The authors take an existing technique and apply it to an existing environment, defining (I believe new) metrics to talk about side effects. The empirical results are reasonable though like much of RL research, it’s very difficult to understand the effects of the many system choices that must be made. For example, the authors note that the high-Z cases result in poorly learned Q functions. This makes it impossible to separate out the effects of higher Z in AUP vs. the approximation effects it causes.

Weaknesses: I don’t find the theoretical results illuminating at all. In the current presentation I don’t see how they provide “intuition about how the AUP penalty term works” nor is it clear how they validate, support, or clearly relate to the experimental results (I will caution that this is the first I have seen the AUP term, so I defer to others who understand the theoretical context).

Correctness: One thing looks very suspicious in the results. The hyperparamter sweep (sec 5.2), for Z=1, N_env=8, lambda=.01. The side effect score is ~0. This is the default configuration which is shown as the AUP line in the upper left chart of Figure 3. That does not look like something that should round to 0. Further, the non-monotonicity of the lamda effect (which increases the strength of the AUP term increases the side effect score is quite surprising. I would like the authors to verify this results and if correct, justify it in the text.

Clarity: There are only minor wording issues; the structure and general writing are well done.

Relation to Prior Work: I am not an expert in this area so can not comment on the larger placement of this work. However, the discussion in Sec 2 reads more as a laundry list of related work without clearly stating how this work addresses limitations or similar ideas, etc.

Reproducibility: No

Additional Feedback: For such a heavily experimental work, I think it’s essential that the code be released, but I see no reference to whether it will be available. The code in supplemental appears to only be for data plotting. For example, the random generation of the environments is not specified in any detail that it could be reproduced.


Review 3

Summary and Contributions: This paper introduces a scalable method for penalizing side effects that successfully avoids side effects in the SafeLife gridworld environments. The method penalizes change in ability to achieve auxiliary reward functions (which can be randomly generated). It is implemented by training a VAE on the environment and using the output of the encoder as an auxiliary reward function. The paper also provides some theoretical results to motivate the design of the penalty.

Strengths: This paper makes significant progress on an important problem in AI safety. This is the first scalable implementation of a side effect penalty that works on complex dynamic gridworlds (avoiding side effects while obtaining full reward). SafeLife is a challenging procedurally generated environment suite designed to test for side effects, including delayed effects (e.g. setting patterns in motion that disrupt other patterns later). Scalability has previously been a concern for side effect penalties, so it's great to see advances in this area.

Weaknesses: It's somewhat unclear how to interpret the theoretical results, in particular what it means for the penalty to be "small" or "large". Theorem 2 suggests that "small" means "at most 1", however it does not show that movement penalties are small in the general (non-deterministic) case, since the upper bound 1/(1-gamma) is at least 20 if gamma > 0.95. Theorem 3 shows that the penalty is bounded from below by a difference in optimal values, but it's not clear why this lower bound is a large number. It would be helpful to give an order of magnitude for this lower bound and compare it to the "small" penalty values in Theorem 2. Overall, this decreases the significance of the theoretical results (though they don't seem like a central point of this paper anyway). Regarding the empirical results, I am not convinced by the claim that the performance degradation as N_env increases is due to the poor generalization of PPO, since PPO does not have a significant increase in side effects or decrease in reward. While this method is a significant step to avoiding side effects in complex environments, it may be difficult to apply to more realistic environments, where there might not be a clear choice of no-op action, and it may not be practical to train a penalty using a long period of random exploration. It would be good for the discussion section to cover these limitations.

Correctness: The empirical methodology seems correct. The proof of Theorem 1 seems correct, though I didn't check Lemma 11. The proof of Theorem 2 is correct (I think line 337 applies Proposition 4 incorrectly, but the resulting statement still holds). The proof of Theorem 3 does not have enough detail to verify correctness - please provide a more detailed proof.

Clarity: The paper is well-written overall, though the theoretical results section is a bit hard to follow. Some notation is not clear, e.g. s_a and s_noop are not defined - is s_a the state reached by following action a from state s, or a point mass distribution at the corresponding state-action pair? There is inconsistent notation between the submission and the supplementary material, e.g. d_AU is defined over states in the paper and seems to be defined over state distributions Delta in the appendix, which is confusing.

Relation to Prior Work: The prior work section covers a wide range of related works, but does not always discuss how this paper differs from them.

Reproducibility: Yes

Additional Feedback: Update after rebuttal: I think the rebuttal addressed the reviewer comments about the experiments well, and I am increasing my score accordingly. I agree with the other reviewers that the theoretical results are not clearly presented and don't add much to the paper in their current form. I think the empirical result is an important novel contribution that stands on its own, and I'd be happy for the theory section to be moved to the appendix to make space for more details on the experiments and connection to related work.


Review 4

Summary and Contributions: The paper presents methodology for scaling Attainable Utility Preservation (AUP) from toy examples to larger games of Conway's Game of Life. The authors show that their method succeeds in minimising side effect occurred by the RL policy while yielding higher reward.

Strengths: Paper presents work in the important area of AI controllability and prevention of undesired behaviour. This is relevant to the NeurIPS community and the broader RL community. Paper provides theoretical grounding for their method, which to the best of my knowledge is correct. Paper evaluates method against baselines and ablations as well as results for different hyperparameter settings. Supplementary contains ample of details regarding, training, derivations and setup.

Weaknesses: Reading the paper without intimate knowledge of the SafeLife environment is difficult. Many different scenarios are references (brief) explanations to what they entail. Paper should more clearly state the contributions compared to previous work and highlight what's new in their work compared to the original work eg in [25]. Interpreting the meaning and impact of the side effect score is difficult. What does it mean that AUP has 46% of the side effects of PPO in append-spawn. How would this translate to more realistic environments?

Correctness: Yes, though the practical significance of having a lower impact score by X% is somewhat unclear. Several of the main results in figure 3 have large overlaps also indicating that while AUP generally outperform PPO and baselines it is unclear how impactful this is.

Clarity: Paper is generallt well written without typos and with easy to follow. As mentioned above the biggest issue is lack of brief explanation of the SafeLife scenarios.

Relation to Prior Work: More clearly presenting what's new compared to [25] would make the paper better.

Reproducibility: Yes

Additional Feedback:


Review 5

Summary and Contributions: This paper extends the Attainable Utility Preservation framework introduced in Turner et al, 2020 to the high-dimensional setting of SafeLife. They show that variants of AUP agents outperform a PPO baseline on different levels of SafeLife in terms of minimizing side effects loss while attaining comparable returns. They find that, in contrast to Turner et al, which used a set of 25 auxiliary reward functions, they achieved best results with a single reward function that was learned as a 1D output of a variational autoencoder trained on 100k exploratory frames.

Strengths: This work provides strong evidence that AUP can be successfully applied to higher-dimensional deep RL settings. AUP is a recent, promising approach to AI safety unifying many previous approaches to learning safe policies that avoid unnecessary side effects, so this result is a useful piece of evidence further endorsing the viability of this approach.

Weaknesses: While the result is interesting, many of the design decisions behind the models and training procedures seemed poorly motivated and discussion on their nuances lacking. - Why represent the rewards as a function of a VAE used to encode exploratory frames in the environment? How does more or less exploration impact the effect of the auxiliary rewards learned this way? - What is the impact of environment ordering during training? Are there any curriculum effects at play? Why train with such a curriculum in the first place? An ablation demonstrating the necessity of this approach would contextualize this decision. - The language stating "the agent learns R_AUP" was confusing (Line 138). The reviewer believes this language was meant to describe "training the Q_i functions" corresponding to each R_i. This confusing language was used in the caption for the reward learning curves in the supplementary materials, which seems supported by the statement starting on Line 202: "In the supplementary material..." - Some key concepts are not defined, for example "initial state reachability" (Line 42). - It seems the claim that safe reinforcement learning focuses on just avoiding negative side effects during training is inaccurate. As stated in the abstract of a paper the authors cite (García et al, 2015), safe RL is also concerned with safety at deployment. - The language is often unclear, e.g. Line 211: "AUP stops moving entirely." The reviewer believes this is referring to the AUP learning a policy in which the agent remains immobile. - The discussion around the theoretical results (3.2) does not add much insight to the experiments and results presented in the paper. - The contribution is not very novel, as it is simply applying AUP as presented in Turner et al, 2020 to another environment, with little to no modification. - Not clear why Lines 55-57 are included in the related work, as they do not seem particularly relevant to safe RL.

Correctness: The experimental results and methodology seem to correctly show that AUP agents perform better in various SafeLife environments in terms of minimizing negative side effects, while reaching comparable performance to PPO. However, some methodological choices are not properly motivated: - Why is the curriculum-based training approach needed vs. training and evaluating on each level in turn? - Why learn the R_i functions as the output of a VAE? - There is no random R_i baseline. - No details of the actual actor-critic model nor full training objective is provided. - Would be valuable to benchmark not just against PPO, but also other SOTA safe RL approaches for higher-dimensional settings, such as the Lagrangian and Constrained Policy Optimization methods presented in Ray et al, 2019. This line of research was also not cited in the prior work.

Clarity: While many concepts are introduced with clarity, the paper uses unclear language in several parts of the paper, as mentioned above. Notably - No details of the actual actor-critic model nor full training objective is provided. - Line 211: "AUP stops moving entirely." The reviewer believes this is referring to the AUP learning a policy in which the agent remains immobile. - Some key concepts are not defined, for example "initial state reachability" (Line 42). - The language stating "the agent learns R_AUP" was confusing (Line 138).

Relation to Prior Work: - Some prior work is included, but not properly contextualized, while other key prior work in safe RL is not mentioned. The work around SafetyGym and CPO presented in Ray et al, 2019 should be included.

Reproducibility: No

Additional Feedback:

[Author Response · NeurIPS 2020]

We thank the reviewers for their feedback. In particular, (R1, R3, R5) asked many specific and thoughtful questions,
with (R1) marking questions by priority. Thank you.

We're glad that all reviewers agree that the paper is well-written and that side effect avoidance is an important AI safety
problem. We are excited that (R1) thinks this work is ground-breaking, novel, and will spur further research, that (R3)
is excited by our scaling of AUP, and that our results are considered strong (R1, R5) and significant (R1, R3).

**Experiments.**   (R1): using multiple auxiliary reward functions performed the same or worse than $|\mathcal{R}| = 1$. (R1, R5)
ask for more comparisons. We will include results for DQN and for additional auxiliary reward functions. At (R1)'s
suggestion, we tried using the primary reward function as the auxiliary reward function. This condition achieved return
and side effects comparable to PPO's, as its attainable utility shifts did not correlate with side effects.

(R5) asks why we didn't compare to state reachability preservation [14] or to AUP with uniformly randomly drawn
auxiliary reward functions over observations (like [25]). Unfortunately, neither approach is remotely viable in SafeLife.
We estimate that there are billions of reachable states in any given SafeLife level. Accordingly, we're aware of a team
trying to train reachability-preserving agents, but even the append-still-easy task was far too hard. If we generated
reward functions by uniformly randomly drawing a reward for every state, the corresponding Q-functions would have
extremely high sample complexity in an environment like SafeLife. We used a VAE because the encoder provided
sufficient structure for quickly learning the auxiliary Q-function.

(R2, R4, R5) ask what distinguishes our work from [25]. The original AUP paper suggested that state reachability
preservation avoids the same breadth of side effects as AUP in [25]'s toy environments. We show that unlike reachability
preservation, AUP scales to SafeLife; as acknowledged by (R1, R3), we are the first to demonstrate compelling results
on any complex side-effect avoidance environment. Furthermore, we showed that AUP competitively accrues reward
while avoiding side effects, while equipped with a single auxiliary reward function which was learned *unsupervised*,
whereas [25] drew several dozen auxiliary reward functions from the uniform distribution.

(R3, R4) wonder about scaling AUP to even more complex environments. We share their interest in this prospect.
Realistic settings might have too many side effect opportunities for a supervised penalty to work well. We believe that
the efficacy of AUP's unsupervised penalty term bodes well for even more challenging domains. (R3): we will add
more discussion of the challenges AUP may face when scaling further, such as our assumption of a no-op action.

(R4) asks what the side effect score (defined lines 115-117) means. Roughly, if AUP halves PPO's side effect score, then
AUP disturbed half as many green cells. Disturbing a patch of green cells corresponds to about 4 additional side effect
score. For a qualitative demonstration of the significance of a $46\%$ side effect score reduction, we refer (R4) to the
attached GIF files for append-spawn.

**Training details.**   (R1): agents were evaluated on their $N_{\mathrm{env}}$ training environments. AUP had no data advantage – it
trained for the standard 5M total steps. While $\mathrm{AUP_{proj}}$ skips the 100K VAE exploration steps, it still learns its auxiliary
Q-function for the first 1M steps. Given more training time, AUP does not overfit and increase side effects. To the
contrary, when running three seeds from 5M steps (the paper's time limit) to 15M steps, AUP's side effect score changed
as follows: append-still-easy: $-24\%$, append-still: $-53\%$, append-spawn: $+17\%$, prune-still-easy:
$-12\%$. We performed hyperparameter search for append-still-easy, and then applied the method successfully to
the other three tasks without further tuning. "Random exploration" refers to a uniformly random exploration policy.

(R2): the hyperparameter sweep shows the average side effect score and episodic return for the *last* episode, while
Figure 3's charts show a rolling average. The discrepancy was unintentional; we agree it is confusing and will fix it.

(R5): we used curriculum training (following the original SafeLife paper, [27]) because PPO fails to learn with just
one environment. "Learning $R_{\mathrm{AUP}}$" refers to the process of training the agent with respect to the AUP reward function
defined in equation 1. Note that when training with respect to $R_{\mathrm{AUP}}$, the auxiliary Q-functions are fixed, as they have
already been trained. We will clarify the difference between auxiliary and AUP training in camera-ready.

**Novelty.**   (R2) finds our work lacks novelty because we "take an existing technique and apply it to an existing
environment". This misses our main contribution: demonstrating AUP's scalability, which is a crucial consideration in
AI safety. As acknowledged by (R1, R3, R5), our work provides the *first* strong empirical evidence that AUP (and side
effect measures more generally) can scale to any kind of complex environment.

**Clarity.**   The prior work section and the SafeLife task explanations will be improved until they are as clear as the rest
of the paper, and we will incorporate (R1, R5)'s suggested citations. (R2): we will make all code available. We are
committed to providing the clearest possible camera-ready paper and look forward to refining the paper to (R1, R2, R3,
R4, R5)'s satisfaction.

[Meta-Review · NeurIPS 2020]

After discussing the paper, all reviewers agree that the empirical results presented in this paper are significant and novel. That said, there are concerns regarding the presentation of the theoretical results, in particular, that they are not clearly presented and do not add much to the paper. During the discussion, it was suggested the theory section to be dropped to make space for more details on the experiments and connection to related work. I encourage the authors to take these recommendations seriously and either address the reviewer concerns regarding the theory section and positioning with respect to related work. I am confident the authors will be able to improve the paper in the remaining time, and recommend acceptance.